# A measurement invariance analysis of selected Opioid Overdose Knowledge Scale (OOKS) items among bystanders and first responders

**James A. Swartz**[1]*, **Qiao Lin**[2], **Yerim Kim**[1]

**1** Jane Addams College of Social Work, University of Illinois Chicago, Chicago, Illinois, United States of America, **2** College of Education, University of Illinois Chicago, Chicago, Illinois, United States of America

☉ These authors contributed equally to this work.
* jaswartz@uic.edu

**Data Availability Statement:** All relevant data are within the paper and its Supporting Information files.

## Abstract

The Opioid Overdose Knowledge Scale (OOKS) is widely used as an adjunct to opioid education and naloxone distribution (OEND) for assessing pre- and post-training knowledge. However, the extent to which the OOKS performs comparably for bystander and first responder groups has not been well determined. We used exploratory structural equation modeling (ESEM) to assess the measurement invariance of an OOKS item subset when used as an OEND training pre-test. We used secondary analysis of pre-test data collected from 446 first responders and 1,349 bystanders (N = 1,795) attending OEND trainings conducted by two county public health departments. Twenty-four items were selected by practitioner/trainer consensus from the original 45-item OOKS instrument with an additional 2 removed owing to low response variation. We used exploratory factor analysis (EFA) followed by ESEM to identify a factor structure, which we assessed for configural, metric, and scalar measurement invariance by participant group using the 22 dichotomous items (correct/incorrect) as factor indicators. EFA identified a 3-factor model consisting of items assessing: basic overdose risk information, signs of an overdose, and rescue procedures/advanced overdose risk information. Model fit by ESEM estimation versus confirmatory factor analysis showed the ESEM model afforded a better fit. Measurement invariance analyses indicated the 3-factor model fit the data across all levels of invariance per standard fit statistic metrics. The reduced set of 22 OOKS items appears to offer comparable measurement of pre-training knowledge on opioid overdose risks, signs of an overdose, and rescue procedures for both bystanders and first responders.

## Background

Although overshadowed the past two years by the rapid, global emergence of COVID-19 as a major public health threat, the decades-long opioid epidemic continues unabated, worsening

**Funding:** JAS recognizes funding from the Illinois Department of Human Services/Division of Substance Use Prevention and Recovery (SUPR) through a grant (#H79 SP022140-01) from the Substance Use and Mental Health Services Administration (SAMHSA) Center for Substance Abuse and Prevention (CSAP). https://www.samhsa.gov/ https://www.dhs.state.il.us/page.aspx?item=29759 The funders had no role in study design, data collection, analysis, decision to publish, or preparation of the manuscript.

**Competing interests:** The authors have declared that no competing interests exist.

in 2020. For the 12-month period ending in May 2020, 81,243 opioid overdose-related fatalities OORF were reported to the CDC, representing the largest 12-month increase in fatalities since 2015 [1]. At least part of the recent trend reversal in OORF is directly related to COVID-19 [2]. The pandemic increased stress, isolation and loneliness, as well as housing instability, and made accessing drug treatment more difficult [3]. Given the enduring nature of the opioid epidemic, however, and despite the lessening of the COVID-19 pandemic over the past year in the US, opioid misuse and OORF remain substantial public health concerns [4].

Prominent among CDC recommendations for reducing OORF is to "expand the provision and use of naloxone and overdose prevention education" with an emphasis on raising awareness about the "critical need for bystanders to have naloxone on hand and use it during an overdose" [1]. Expanding community naloxone access has been one of three US Department of Health and Human Services priority areas for addressing the opioid crisis [5].

Providing take-home naloxone (THN), a competitive opioid antagonist that can rapidly reverse the life-threatening effects of an opioid-related overdose, to non-medically trained "bystanders" (e.g., a heterogeneous group that can include opioid users, their social networks and family members, and social service staff) has become an important component of expanding naloxone access [6–8]. THN has become more widespread over the past 5 years, abetted by the passage of "Good Samaritan" laws; increased availability through pharmacies, hospitals, and emergency departments; Medicaid expansion under the Affordable Care Act; and the availability of a nasal spray, obviating the need for administration by injection [9–11]. Despite increased availability, there continues to be a need for further expansion of community-based overdose education and naloxone distribution (OEND) for bystanders as well as for non-medical first responders such as the police.

Though studies of whether THN is a moral hazard that increases risky opioid use (i.e., risk compensation) have yielded mixed results, the preponderance of the evidence suggests that, on balance, THN reduces OORF [5,12,13]. Moreover, while it is possible the effectiveness of providing THN is enhanced when combined with education and training on recognizing the signs of an overdose and how to correctly administer naloxone, current evidence that providing education and training provides additional benefit has not been well established [5,13–18].

As the continuation of THN programs that incorporate education and training is likely and because there is no single standardized set of instructional materials or manualized training protocol, it is important to have validated instruments for assessing whether any specific training effectively increases basic knowledge on recognizing the signs of an opioid overdose and how to correctly administer naloxone for the intended audience, whether bystanders or first responders. Having valid information on whether education and training effectively increase knowledge and skills are necessary pre-requisites to determining if OEND further contributes to reductions in OORF beyond naloxone distribution alone and for whom and for determining which training content and materials are most effective.

The Opioid Overdose Knowledge Scale (OOKS) was the first validated instrument of which we are aware to be used in conjunction with OEND [19] although other instruments have become available more recently [20]. Consisting of 45 items in its original version, the OOKS continues to be frequently cited in studies that evaluate THN/OEND trainings [21–23]. The OOKS has been translated into several languages for use in European countries and a short, 10-item form (i.e., the Brief Opioid Overdose Knowledge Scale [BOOK]) has been developed as well as a version adapted for use to assess knowledge specific to prescription opioids [24–26].

Despite its popularity, there remain questions about whether the OOKS is optimally constructed for use with bystanders as well as first responders. In the original study, the OOKS was administered to both bystanders (i.e., small samples of friends and family members of

heroin users; N = 42) as well as healthcare professionals (N = 56). The BOOK was developed based on larger samples of illicit opioid users and patients prescribed an opioid for chronic pain treatment [N = 848; Dunn, Barrett [24]]. In both studies, most participants would be considered bystanders with the status of the trained healthcare professionals unclear. Police, who are very often the first responders when overdoses occur, were not involved in the development of the OOKS or the BOOK.

Considering the continuing need to assess the effectiveness of different OEND training strategies for groups of participants with varying knowledge and professional training, this study sought to evaluate the performance of a selected subset of OOKS items when used as a training pre-test for first responders as well as bystanders. To the best of our knowledge, there has not been a measurement invariance analysis of the OOKS to assess whether it measures basic pre-training knowledge for these two groups for whom naloxone education and training are commonly conducted. To address this gap, we conducted a multigroup analysis of OOKS item equivalence by assessing levels of measurement invariance (i.e., configural, metric, and scalar) for first responders and bystanders. If the OOKS demonstrates measurement invariance, then one version would be suitable for use with both groups whereas lack of measurement invariance suggests either different versions of the OOKS containing different item subsets or different scoring thresholds for bystanders and first responders would be more appropriate. We also wanted to re-examine the OOKS item factor structure using a relatively new analytic tool, exploratory structural equation modeling (ESEM), which combines elements of exploratory factor analysis and confirmatory factor analysis [27,28]. We describe the potential advantages of ESEM in the analysis section below.

## Methods

This study was a secondary analysis of de-identified pre-test data collected as part of an evaluation of a project to reduce OORF through increasing THN availability and conducting OEND trainings in Illinois communities with high OORF rates. The University of Illinois Chicago IRB determined the study did not constitute human subjects research and granted an exemption. This determination was made on the basis of no interaction between the investigators and study participants as well as the analyses would be restricted to fully de-identified data. Given the study's exempt status, the requirement for obtaining written or verbal consent were also waived by the IRB.

### Study setting and sample

The Substance Use and Mental Health Services Administration (SAMSHA) provided funding for a five-year project (2016–2021) to reduce OORF among users of any type of opioid including street drugs such as heroin or fentanyl. Group trainings on naloxone administration and recognizing an overdose were scheduled as needed at each of the six participating sites based on outreach efforts in participating communities. As part of the project evaluation, two sites agreed to administer pre- and post-test quizzes to assess knowledge gain and assess training effectiveness. A 24-item version of the OOKS was used and required about 10 minutes to administer. One site trained both first responders (N = 498) and bystanders (N = 506) whereas the second site exclusively trained bystanders (N = 1,137). We eliminated data from participants with more than 3 missing responses or with a response pattern indicating rote responding such as selecting the same response option across all items (N = 346, 16.2%). This yielded a final analytic sample of 1,795 participants composed of 446 (24.8%) first responders and 1,349 (75.2%) bystanders.

First responders were composed predominantly of police whereas composition of the bystander groups varied considerably. Aggregate data provided by the site that conducted bystander trainings indicate the following types of bystanders participated: active drug users/ syringe exchange program clients; family members and friends of drug users; staff working in substance use treatment programs; staff in human services agencies such as shelters and transitional living programs; and staff working in public institutions such as schools, restaurants, and libraries. In total, the two participating sites conducted 109 trainings between March 10, 2018 and January 20, 2020. The mean number of participants per training was 36.1 (sd = 25.3), with the size of bystander trainings larger and more variable (mean = 38.7, sd = 27.1) compared with first responder trainings (mean = 24.9, sd = 9.8; $t_{(df = 2,029) =}$ 10.0, p < .001).

We did not collect individual demographic information to preserve training participant anonymity. However, aggregated data indicate the following demographic composition of the study sample: most participants were male (80.1%); white (82.1%); and between the ages of 25 to 44 (57.2%) with smaller proportions reporting they were 18 to 24 years old (19.4%) or 45 to 64 years old (10.6%). About 8.3% were African American/Black with 12.7% indicating Latino/ Latinx ethnicity.

## Statistical methods

**Variables.** Because training content and delivery varied by trainer, setting, and participant composition, we restricted our analyses to the pre-tests to eliminate variance owing to training-related factors that could affect post-test responses. At project outset, a panel composed of site directors and training practitioners reviewed the original OOKS for use in the evaluation of trainings. Because of the limited time for trainings, 50 to 60 minutes at most, the review panel shortened the 45-item OOKS by removing 21 items they believed were redundant or less informative to the project training goals. After preliminary analyses of the collected training data, we removed two additional items–one should call an ambulance when managing an overdose as well as stay with the person until an ambulance arrives—that were answered correctly by 98% or more of all participants in each group and consequently lacked variability, causing convergence problems during the measurement invariance analyses. The resulting 22-items for the modified version of the OOKS are shown in Table 1 with the original OOKS section and item number provided parenthetically.

The pre-test was administered as a self-report paper questionnaire. For each item, participants were asked to indicate true, false, or unsure/unknown. The self-reported data were then entered into REDCap [29], downloaded and scored as correct or incorrect according to the answer key provided with the original OOKS [19]. All unsure/don't know responses were coded as incorrect, effectively converting the items from trichotomous to dichotomous.

## Measurement invariance analysis

We used Stata version 17.0 for data screening and generating bivariate statistics [30]. We first ran descriptive analyses to assess percentage correct by item as well as number of total correct responses, disaggregated into three groups: bystanders trained at site A; first responders trained at site A; and bystanders trained at site B. For each item, a bivariate logistic regression was used to compare the odds of a correct response by group. We also calculated a mean score for each group based on the total correct and compared these using a one-way ANOVA. To assess the 22 OOKS items for measurement invariance, we first ran an exploratory factor analysis (EFA) to determine the number of factors for the pre-test instrument. Although a factor structure has been identified for the original 45-item OOKS [19], we wanted to re-assess this given we were using an item subset. We examined factor structure in several ways: using

**Table 1. Percentage correct responses to selected OOKS items by OEND training group.**

| Selected OOKS Pre-test Items | Site A—First Responders | Site A—Bystanders | Site B—Bystanders | Total | | Sig | Sig |
|---|---|---|---|---|---|---|---|
| | (N = 446) | (N = 412) | (N = 937) | (N = 1,795) | | (FR—BYS) | (BYS_A—BYS_B) |
| **Which of the following factors increase the risk of a heroin (opioid) overdose?** | Correct % | Correct % | Correct % | Correct % | | | |
| Switching from smoking to injecting heroin (A2) | 82.7 | 77.5 | 77.9 | 79.0 | % | NS | NS |
| Using heroin with other substances, such as alcohol or sleeping pills (A3) | 96.6 | 95.6 | 97.6 | 96.9 | | NS | NS |
| Increase in heroin purity (A4) | 95.1 | 94.4 | 95.1 | 94.9 | | NS | NS |
| Using heroin again soon after release from prison (A8) | 85.0 | 77.0 | 75.0 | 78.0 | | *** | NS |
| Using heroin again after a detoxification treatment (A9) | 90.4 | 85.0 | 84.0 | 85.8 | | ** | NS |
| **Which of the following are indicators of an opioid overdose?** | | | | | | | |
| Having blood-shot eyes (B1) | 39.9 | 27.4 | 30.4 | 32.1 | | *** | NS |
| Slow or shallow breathing (B2) | 95.1 | 88.1 | 84.2 | 87.8 | | *** | NS |
| Lips, hands or feet turning blue (B3) | 89.0 | 78.9 | 79.8 | 81.9 | | *** | NS |
| Loss of consciousness (B4) | 97.3 | 94.0 | 91.1 | 93.3 | | *** | NS |
| Deep snoring (B7) | 66.8 | 55.2 | 49.6 | 55.2 | | *** | NS |
| Very small pupils (B8) | 82.3 | 73.6 | 69.9 | 73.8 | | *** | NS |
| Agitated behaviour (B9) | 47.3 | 38.7 | 33.2 | 38.0 | | *** | NS |
| Rapid heartbeat (B10) | 46.0 | 33.4 | 29.6 | 34.5 | | *** | NS |
| **Which of the following should be done when managing a heroin (opioid) overdose?** | | | | | | | |
| †Call an ambulance (C1) | 99.8 | 98.1 | 97.1 | 97.9 | | ** | NS |
| †Stay with the person untill an ambulance arrives (C2) | 99.3 | 97.3 | 98.8 | 98.6 | | NS | NS |
| Give stimulants (e.g. cocaine or black coffee) (C5) | 56.1 | 50.4 | 52.6 | 53.0 | | NS | NS |
| Place the person in the recovery position (on their side with mouth clear) (C6) | 92.8 | 80.2 | 79.6 | 83.0 | | *** | NS |
| Put the person in bed to sleep it off (C11) | 69.5 | 54.2 | 55.6 | 58.7 | | *** | NS |
| **What is naloxone used for?** | | | | | | | |
| To reverse the effects of any overdose (D4) | 63.9 | 46.5 | 38.2 | 46.5 | | *** | ** |
| **How can naloxone be administerd?** | | | | | | | |
| Into mouth or swallowed orally (E4) | 79.4 | 63.0 | 45.8 | 58.1 | | *** | *** |
| **How long do the effects of naloxone last?** | | | | | | | |
| 2 to 6 hours (H3) | 27.3 | 15.7 | 21.1 | 21.4 | | *** | NS |
| **Please indicate which of the following statements are correct** | | | | | | | |
| If the first dose of naloxone has no effect a second dose can be given (I1) | 72.2 | 52.3 | 49.8 | 56.0 | | *** | NS |
| Someone can overdose again even after having received naloxone (I3) | 79.6 | 72.6 | 70.2 | 73.1 | | *** | NS |
| Naloxone can provoke withdrawal symptoms (I6) | 50.2 | 43.8 | 40.6 | 43.7 | | ** | NS |

**Note.** All figures shown are percentages of correct responses for each item by participant group and site. The labels in parentheses by each item reference the original OOKS instrument item section and number. All significance tests are based on Pearson chi-square tests with 2 degrees of freedom. Only results significant at $p < .01$ are reported. The first set of significance tests compare the comined set of bystanders with first responders. The second set compares the percent of correct responses for bystanders at site A with bystanders at site B.

†These two items, "Call an ambulance" and "Stay with the person until an ambulance arrives", were removed from the final version given the very high correct response rate across all participants and the resulting convergence issues caused when these items were included in the modeling steps.

** = $p < .01$

*** = $p < .001$, NS = Non-significant.

principal factor analysis with goemin rotation and robust weighted least squares in Mplus version 8.7 [31] to generate models with 2 to 6 factors and comparing the CFI, TLI, RMSEA, and SRMR fit statistics for each model. We also conducted parallel analysis in the R software program version 4.1.3 [32] using the fa.parallel function in the psych package for R [33] assessing polychoric correlations, given the factor indicators were binary (correct/incorrect). Based on analyses of the relative accuracy of different metrics for most accurately determining the number of factors in an EFA under varying circumstances (e.g., sample size, underlying number of factors, factor correlations and loading), we gave greater weight to the RMSEA statistic [34] in selecting a final model. Our goal was to find the model with the minimum number of factors that still provided adequate fit to the data.

We then used exploratory structural equation modeling (ESEM) to test levels of measurement invariance for the selected factor model. ESEM combines EFA, and CFA/SEM and is less restrictive than CFA because factor cross-loadings are not constrained to zero. This potentially permits estimation of better-fitting and more realistic models whereby items can be associated with multiple factors [27,35]. We directly assessed this by estimating two models based on CFA–one that did and one that did not allow for selected residual covariances–with two corresponding models based on ESEM. We used conventional fit statistics and thresholds to evaluate model fit [36]: root mean square error of approximation (RMSEA) $\leq 0.5$; comparative fit (CFI) and Tucker–Lewis indices (TLI) $\geq 0.90$; and standardized root mean square residual (SRMR) $\leq 0.8$.

We assessed the OOKS pre-test for measurement invariance for first responders and bystanders across three increasingly restrictive measurement levels using the best fitting model derived from the CFA-ESEM comparative analyses: configural, metric or "weak invariance", and scalar [37]. Briefly, configural invariance is indicated when factors are composed of the same items for each group, but the item factor loadings and intercepts/thresholds are allowed to vary across groups; with metric invariance, an equality constraint is applied to the factor loadings of each item to hold them equal across groups; and finally, with scalar invariance, an additional equality constraint is applied to the item intercepts/thresholds. For each successive model, we again used a conventional set of fit statistics and thresholds to evaluate model fit.

## Results

### Descriptive statistics

Table 1 shows the percentage of total correct responses by item as well as by participant group. In general, the percentages of correct answers were similar among the two bystander groups with only two of the more difficult items–whether naloxone can be used to reverse an overdose for *any* drug and how long the effects of naloxone last–showing significantly different proportions of correct responses. There were many more significant differences between the combined bystander groups and the first responders with first responders having a higher proportion of correct responses for 18 of the 22 OOKS items. Across all participants, the three items that were answered correctly least often were the length of time naloxone lasts (21.4% correct), having blood shot eyes (32.1% correct), or displaying agitated behavior (38.0% correct) as indicators of an opioid overdose.

On average, first responders answered 16 of the 22 items correctly (mean = 16.1, SD = 3.5) whereas bystanders at site A averaged under 14 correct responses (mean = 13.9, SD = 3.9) as did bystanders at site B (mean = 13.5, SD = 4.2). These overall differences were statistically significant ($F_{(2,1792)} = 63.82$, $p < .001$) with post-hoc analyses indicating that the first responders scored significantly higher than either bystander group but that the small difference among the bystander groups was non-significant.

## Exploratory factor analyses

Results for the exploratory analyses for the 2 to 6 factor models ruled out only the 2-factor model as not providing adequate fit (RMSEA = .053 [95% CI = .050 - .056]; CFI = .905; TLI = .883; SRMR = .077). Models with 3 to 6 factors fit the data well per these same statistics with each showing improvement in model fit over the preceding model as determined by Chi-square tests comparing each model with the model having k-1 factors. These comparisons supported the 6-factor model as providing the best fit as did inspection of the scree plot obtained following parallel analysis. Despite these results, and for several reasons enumerated below, we selected between the 3- (RMSEA = .044 [95% CI = .041 - .048]; CFI = .941; TLI = .919; SRMR = .067) and 4-factor models (RMSEA = .044 [95% CI = .034 - .041]; CFI = .962; TLI = .941; SRMR = .057), ultimately selecting the 3-factor model for further analysis.

As noted by Finch (2020) in his simulation study, when the fit statistics are inconsistent with the underlying factor structure of the simulated models, the statistics tend to favor over-factored (i.e., too many factors) results. Parsimony was also an important consideration. All else being equal, simpler models are better given relatively similar fit statistics. One of the factors in the 4-factor model had only two items with loadings greater than .50, with the 5 and 6-factor models having a similar factor structure whereby only several items had loadings greater than .50. Current recommendations are that a factor is identified when there are 3 or more items with sufficient loadings [38]. Finally, in subsequent preliminary analyses, the 4-factor model failed to converge at the metric invariance step suggesting it could be overly complex. Given that both the 3- and 4-factor model fit statistics indicated both fit the data well and the item to factor structure appeared to be more robust in the 3-factor model, which had more than 2 items with loadings > .50 on every factor, and the pattern of factor loadings for the items made sense substantively, we selected the 3-factor model for measurement invariance testing.

## ESEM model

The final ESEM factor structure for the 3-factor model identified in the EFA step is shown in Table 2, which displays factor loadings and significance levels of the selected OOKS items. Based on the significant factor loadings, we assessed factor 1 (**overdose risks**) as determined by correct identification of opioid overdose risk items drawn from section A of the OOKS. Items corresponding to recognition of overdose signs and drawn from section B of the OOKS were most strongly associated with factor 2 (**overdose signs**). Factor 3 (**rescue/advanced knowledge**) was composed of knowledge related to what to do to rescue a person from an opioid overdose as well as some items corresponding to more advanced knowledge (as reflected by more difficult items on the pre-test) such as whether blood shot eyes or agitated behavior indicate an opioid overdose or how long the effects of naloxone last. Items loading on factor 3 were drawn from across multiple sections (B–I) of the original OOKS.

## CFA and ESEM model estimation comparison

To assess whether allowing estimation of factor cross-loadings in ESEM provided better model fit than constraining the cross-loadings to zero as in CFA, we estimated both CFA and ESEM models with and without allowing 3 selected correlated residual terms that modification indices suggested would improve model fit: OOKS items B10 (rapid heartbeat) and B9 (agitated behavior) as overdose indicators; items C11 (put person in bed to "sleep it off") and C5 (place person in recovery position); and items I6 (provoke withdrawal symptoms) and I3 (someone can overdose again after receiving naloxone). The results of these analyses are shown in Table 3 and provide support for ESEM estimation as resulting in a better fitting model

**Table 2. ESEM three-factor model structure for selected OOKS pre-test items.**

| Which of the following factors increase the risk of a heroin (opioid) overdose? | Factor 1 Overdose Risks | Sig | Factor 2 Overdose Signs | Sig | Factor 3 Rescue/Advanced Knowledge | Sig |
|---|---|---|---|---|---|---|
| Switching from smoking to injecting heroin (A2) | 0.609 | *** | 0.193 | ** | -0.023 | |
| Using heroin with other substances, such as alcohol or sleeping pills (A3) | 0.555 | *** | 0.191 | | -0.007 | |
| Increase in heroin purity (A4) | 0.624 | *** | 0.118 | | 0.099 | |
| Using heroin again soon after release from prison (A8) | 0.490 | *** | 0.264 | *** | 0.074 | |
| Using heroin again after a detoxification treatment (A9) | 0.516 | *** | 0.146 | * | 0.172 | *** |
| **Which of the following are indicators of an opioid overdose?** | | | | | | |
| Having blood-shot eyes (B1) | -0.398 | *** | 0.016 | | 0.673 | *** |
| Slow or shallow breathing (B2) | -0.040 | | 0.815 | *** | 0.031 | |
| Lips, hands or feet turning blue (B3) | 0.017 | | 0.767 | *** | -0.015 | |
| Loss of consciousness (B4) | 0.235 | *** | 0.544 | *** | 0.034 | |
| Deep snoring (B7) | 0.208 | *** | 0.510 | *** | 0.121 | ** |
| Very small pupils (B8) | 0.143 | ** | 0.423 | *** | -0.036 | |
| Agitated behaviour (B9) | -0.288 | *** | 0.061 | | 0.603 | *** |
| Rapid heartbeat (B10) | -0.198 | *** | 0.166 | ** | 0.593 | *** |
| **Which of the following should be done when managing a heroin (opioid) overdose?** | | | | | | |
| Give stimulants (e.g. cocaine or black coffee) (C5) | -0.211 | *** | -0.016 | | 0.471 | *** |
| Place the person in the recovery position (on their side with mouth clear) (C6) | 0.286 | *** | 0.210 | ** | 0.129 | * |
| Put the person in bed to sleep it off (C11) | -0.213 | *** | 0.143 | * | 0.391 | *** |
| **What is naloxone used for?** | | | | | | |
| To reverse the effects of any overdose (D4) | 0.053 | | 0.030 | | 0.654 | *** |
| **How can naloxone be administerd?** | | | | | | |
| Into mouth or swallowed orally (E4) | 0.038 | | 0.138 | * | 0.607 | *** |
| **How long do the effects of naloxone last?** | | | | | | |
| 2 to 6 hours (H3) | 0.095 | | -0.104 | | 0.691 | *** |
| **Please indicate which of the following statements are correct** | | | | | | |
| If the first dose of naloxone has no effect a second dose can be given (I1) | 0.159 | *** | 0.062 | | 0.626 | *** |
| Someone can overdose again even after having received naloxone (I3) | 0.413 | *** | -0.158 | ** | 0.526 | *** |
| Naloxone can provoke withdrawal symptoms (I6) | 0.320 | *** | -0.167 | ** | 0.588 | *** |

Note. All figures shown are standardized factor loadings on each item based on exploratory stuctural equation modeling with target rotation for a 3-factor model. Model parameters were estimated using weighted least square with means and variance adjusted estimator. The labels in parentheses by each item reference the original OOKS instrument section and item number. Shaded items indicate the highest factor loading for that item.

* = p < .05

** = p < .01

*** = p < .001.

(RMSEA = .032 [95% CI = .028 - .035]; CFI = .971; TLI = .959; SRMR = .049) compared with the best fitting CFA model (RMSEA = .043 [95% CI = .040 - .046]; CFI = .934; TLI = .924; SRMR = .072).

## Measurement invariance analysis

Our final analyses tested whether the factor structure for the best fitting ESEM model (shown in Table 2) fit the data equally well for bystanders and first responders or whether a different factor structure would better fit the data for each group. The results of these measurement invariance analyses are also shown in Table 3. We began by estimating the configural model,

**Table 3. Exploratory models, parameter constraints, and fit statistics for increasingly restrictive measurement invariance models.**

| | Model Parameters | | | Fit Statistics | | | | |
|---|---|---|---|---|---|---|---|---|
| | Factor Loadings | Item Thresholds | Residual Covariances | Chi-2 (df) | TLI | CFI | RMSEA (95% CI) | SRMR |
| **Exploratory Models (No Grouping)** | | | | | | | | |
| CFA—Uncorrelated Residuals | Constrained | NA | None | 1598.967 (206) | 0.845 | 0.861 | .061 (.059 - .064) | 0.087 |
| CFA—Correlated Residuals | Constrained | NA | Included | 822.203 (203) | 0.924 | 0.934 | .043 (.040 - .046) | 0.072 |
| ESEM—Uncorrelated Residuals | Targeted | NA | None | 762.781 (168) | 0.919 | 0.941 | .044 (.041 - .048) | 0.062 |
| ESEM—Correlated Residuals | Targeted | NA | Included | 459.627 (165) | 0.959 | 0.971 | .032 (.028 - .035) | 0.049 |
| **Multi-group Measurement Invariance Models** | | | | | | | | |
| ESEM—Configural Invariance (Correlated Residuals) | Free | Free | Included | 602.752 (330) | 0.952 | 0.966 | .030 (.026 - .034) | 0.061 |
| ESEM—Metric Invariance (Correlated Residuals) | Fixed | Free | Included | 593.864 (387) | 0.969 | 0.974 | .024 (.020 - .028) | 0.072 |
| ESEM—Scalar Invariance (Correlated Residuals) | Fixed | Fixed | Included | 654.755 (406) | 0.950 | 0.969 | .026 (.022 - .030) | 0.073 |

Note. All measurement invariance modeling was conducted using Exploratory Structural Equation Modeling (ESEM) with targeted rotation and weighted least squares mean and variance adjusted robust estimation using Mplus (Muthén, & Muthén, 2021). 'Free' means the model parameter was allowed to vary across groups, whereas fixed means the parameter was constrained to be equal across groups. Models where residual covariances were included allowed estimation of the residual covariances between three sets of OOKS items as identified in the text. Otherwise, residual covariances were fixed to zero. In the CFA models, cross-factor loadings were constained to be zero whereas in the ESEM exploratory and measurement invariance models, cross-factor loadings were estimated to be as close to zero as possible.
CFI = Comparative Fit Index; TLI = Tucker Lewis Index; RMSEA = Root Mean Square Error of Approximation; and SRMR = Standardized Root Mean Square Residual.

which has the same overall factor structure but allows the item factor loadings and thresholds to vary across groups. This model fit the data reasonably well (RMSEA = .041 [95% CI = .038 - .045]; CFI = .935; TLI = .911; SRMR = .071). The metric model showed improved fit (RMSEA = .024 [95% CI = .020 - .028]; CFI = .974; TLI = .969; SRMR = .072) relative to the configural invariance model. The scalar invariance model fit statistics showed only slightly poorer but still very good model fit compared with the metric invariance model. Moreover, per thresholds recommended by Finch (e.g., change in CFI > = .01, change in SRMR > .03, change in RMSEA > = .015) for comparing fit differences between models where differences greater than the thresholds indicate poorer fit, differences between the scalar and metric invariance models were well below the recommended thresholds. We therefore concluded that the OOKS items met the criteria for establishing scalar invariance for first responders and bystanders.

## Discussion

We believe the findings from this study have implications for assessing participants attending OEND trainings as well as for anticipating areas of training that might require more emphasis, especially among bystander trainees. From a measurement standpoint, we found that a 3-factor model adequately represented the variances and covariances among the selected OOKS items.

Although our goal was not to develop a shortened version of the OOKS given the existence of the 12-item BOOK, the study that developed the BOOK also found a corresponding though not entirely overlapping 3-factor structure: opioid knowledge, opioid overdose knowledge,

and opioid overdose response knowledge [24]. The original OOKS was organized into four sections based on substantive considerations that were not derived through statistical analyses [19]. The four sections include: (1) questions on knowledge of risks for an overdose; (2) signs of an overdose; (3) actions to take in response to an overdose; and (4) the correct use of naloxone. Our 3-factor structure more closely approximates the BOOK factor structure whereby, essentially, actions to take in response to an opioid-related overdose and the correct use of naloxone form a single rather than two separate factors. We believe the similarity of the factor analytic results across our study and the BOOK development study as well as the general consistency of both factor-analytic studies with the original substantively driven design of the original OOKS suggests the 3-factor structure we identified is robust and well captures the knowledge areas that should be covered during OEND trainings.

Our analyses then focused on determining if the selected set of OOKS items met criteria for measurement invariance among first responders and bystanders. The results supported use of this subset of OOKS items for assessing pre-training knowledge among both groups of participants. The results also supported using the items for comparative purposes given the 3-factor model evidenced scalar invariance. This indicates that comparisons of the mean scores for these two groups of OEND participants are valid. Further work would need to establish fully whether these findings are applicable to the BOOK short-form and the OOKS parent instrument, but our findings suggest the original OOKS and/or an item subset such as with the instrument used in this study have broad applicability for use in OEND training assessment.

From a practical standpoint, the results indicated that first responders are likely to have greater knowledge of the risks for an opioid-related overdose, signs of an overdose, and what to do to reverse an overdose prior to training compared with bystanders. That this finding held across different groups of bystanders trained in two different counties, with considerable heterogeneity within each bystander group, gives us some though not complete confidence in the generalizability of the findings.

Whereas trainings for both groups should address all necessary knowledge relevant to recognizing and reducing opioid-related overdoses and fatalities, trainings for first responders might focus more on details such as exactly how long naloxone can be expected to last and which symptoms do not specifically indicate an opioid-related overdose (e.g., having bloodshot eyes, agitated behavior, and a rapid heartbeat). These were the issues generating the highest proportion of incorrect responses among first responders. Conversely, bystanders tended to begin trainings without knowledge in these same areas but also lacked information more generally such as whether naloxone can be used to reverse overdoses for any drug. If possible, longer OEND trainings might be needed to address fully the greater background knowledge deficit among bystanders relative to first responders.

## Limitations

Our determination to seek a model with the smallest number of factors to represent the OOKS data means there is a possibility that our model is "under-factored", and that the data might be better represented by a model with more factors. However, we assessed models with more factors and did not find they provided substantially better fit, had issues with model estimation during the measurement invariance testing, and the additional factors included had only several items with loadings greater than .50. The models with more factors did not seem to improve either fit or substantive interpretation. In addition, although the collaborating sites carefully deliberated on which items to remove to avoid redundancy, it is possible that better performing items were remove or the original OOKS item set could have demonstrated better

or worse measurement invariance. For this reason, we recommend a replication of this study on the full OOKS item set.

Generalizability of study findings remains a potential issue. Although we had a large and diverse sample, participants were drawn from two counties in a single midwestern state in the United States and might not be representative of OEND training participants in other parts of the country or in other countries or with other backgrounds. Racial and ethnic minorities were also under-represented in our samples. Moreover, the bystander group was heterogeneous and composed of people who use drugs, family members, professionals working in treatment settings, and so on. Given the data were collected to preserve participant anonymity fully, we could not subdivide this sample to compare these different subpopulations for OEND knowledge and knowledge gaps. Consequently, we were unable to determine if gaps in knowledge among bystanders were differentially attributable to drug users, family members (of drug users), or professionals working in treatment settings. Future research should examine whether this shortened version of the OOKs works equally well for these different subgroups of bystanders.

Last, we also do not know how applicable the findings are with respect to post-training measurement. Clearly, an important goal of administering pre- and post-tests at trainings, is to assess training effectiveness through knowledge gain. It would therefore seem important as a next step, to assess measurement invariance across occasions rather than types of participants. This would help determine if comparing mean pre- and post-training factor scores is statistically valid [39]. We also believe because of the high number of items with significant cross-loadings in the final ESEM model, a bi-factor model with a single factor representing general knowledge across items and a 3-factor representation of specific knowledge as revealed in this study is worth further exploration [40].

## Conclusion

This study supports use of a subset of items of the OOKS and, by inference, the full OOKS or shorter version BOOK, as adjuncts to assess pre-training knowledge among broadly constituted groups of OEND participants. We found that the OOKS items demonstrated scalar measurement invariance supporting the use of scale score comparisons of opioid overdose related knowledge among bystanders and first responders as valid. Although not a focus of the study, we also found that bystanders tend to have larger knowledge gaps in key areas related to recognizing and reversing an opioid overdose, particularly with respect to details on naloxone use and duration of effect. These areas could be more emphasized in future trainings to address these knowledge gaps.

## Supporting information

**S1 Dataset.**
(XLS)

## Acknowledgments

We thank Lauren Cox for her assistance in assembling the manuscript; Sharuti Madan for her help with obtaining and reviewing the background literature; and Kelly Horn for entering the study data. We also want to thank the support of the project site directors and staff that afforded access to and collected the training data for this study: Mila Tsigalis, Kathleen Burke, Rabia Mukhtar, and Scott Kaufman.

## Author Contributions

**Conceptualization:** James A. Swartz, Qiao Lin, Yerim Kim.

**Formal analysis:** James A. Swartz, Qiao Lin, Yerim Kim.

**Methodology:** James A. Swartz.

**Writing – original draft:** James A. Swartz, Qiao Lin.

**Writing – review & editing:** James A. Swartz, Yerim Kim.

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
