## [Decision Letter · Decision Letter 0]

14 Sep 2022

PONE-D-22-18050A measurement invariance analysis of selected opioid overdose knowledge scale (OOKS) items among bystanders and first respondersPLOS ONE

Dear Dr. Swartz,

Thank you for submitting your manuscript to PLOS ONE. After careful consideration, we feel that it has merit but does not fully meet PLOS ONE’s publication criteria as it currently stands. Therefore, we invite you to submit a revised version of the manuscript that addresses the points raised during the review process.

We look forward to receiving your revised manuscript.

Kind regards,

Michelle Melgarejo da Rosa

Academic Editor

PLOS ONE

Journal Requirements:

Additional Editor Comments:

General points must be addressed:

1) In table 1, the visualization of the original data must be available to see all the people's choices. The original questionnaire must be included.

2)The statistics analysis methods must be more detailed, as much as, the comparison criteria among groups.

3) The innovative aspect of the article is not clear. Also, it is not clear the hypothesis of the authors.

4) On page 18 the authors state: "Despite these results, and for several reasons.... What are the several reasons mentioned?

5) There is a lack of literature discussion. The authors do not compare original data with general knowledge. The factors included and the importance of each must be contextualized.

6) The results are not overall representative to make a general statement of it

7) The bystander group of analysis should be divided into drug users, family members, and professionals. The way the authors represent does not point to the right source of misinformation. It is not clear where/who the questionnaire requires modification.

Reviewers' comments:

Reviewer's Responses to Questions

**Comments to the Author**

1. Is the manuscript technically sound, and do the data support the conclusions?

Reviewer #1: Yes

Reviewer #2: Yes

2. Has the statistical analysis been performed appropriately and rigorously? 

Reviewer #1: I Don't Know

Reviewer #2: I Don't Know

3. Have the authors made all data underlying the findings in their manuscript fully available?

Reviewer #1: Yes

Reviewer #2: No

4. Is the manuscript presented in an intelligible fashion and written in standard English?

Reviewer #1: Yes

Reviewer #2: Yes

5. Review Comments to the Author

Reviewer #1: Page 18, first sentence

"From a practical standpoint, the results indicated that first responders are likely to have

greater knowledge of the risks for an opioid-related overdose, signs of an overdose, and what

to do to reverse an overdose prior to training compared with first responders"

I do apologize but it is unclear for me, first responders compared with first responders???

Reviewer #2: General points must be addressed:

1) In table 1, the visualization of the original data must be available to see all the people's choices. The original questionnaire must be included.

2)The statistics analysis methods must be more detailed, as much as, the comparison criteria among groups.

3) The innovative aspect of the article is not clear. Also, it is not clear the hypothesis of the authors.

4) On page 18 the authors state: "Despite these results, and for several reasons.... What are the several reasons mentioned?

5) There is a lack of literature discussion. The authors do not compare original data with general knowledge. The factors included and the importance of each must be contextualized.

6) The results are not overall representative to make a general statement of it.

7) The bystander group of analysis should be divided into drug users, family members, and professionals. The way the authors represent does not point to the right source of misinformation. It is not clear where/who the questionnaire requires modification.

6. PLOS authors have the option to publish the peer review history of their article (what does this mean?). If published, this will include your full peer review and any attached files.

Reviewer #1: No

Reviewer #2: No

---

## [Author Response · Author response to Decision Letter 0]

26 Sep 2022

I have provided all of my responses in the cover letter. But to be certain, here are those responses again:

Dr. Michelle Melgarejo da Rosa

Academic Editor 

PLOS ONE

Re: Manuscript revision and resubmission

Dear Dr. Melgarejo da Rosa: 

Thank you for your consideration of and feedback on the submitted manuscript titled: A Measurement invariance analysis of selected Opioid Overdose Knowledge Scale (OOKS) items among bystanders and first responders. On behalf of my co-authors and myself, we wish to express our appreciation for your and the reviewers’ suggestions for strengthening the manuscript. In this rebuttal letter, we provide detail on the changes made or, in a very few instances, an explanation for why we chose not to make a change based on the critique provided in your notification email. For reference, we have included the provided critiques and then, below each, indicate our response. We have also reviewed the journal’s formatting requirements as provided in the response letter and have tried to ensure that we follow them correctly. Last, in reviewing the manuscript, we caught a few typos and have corrected these as well as updated the background section (in one sentence) to indicate that opioid-related fatalities continue to be an issue despite the waning COVID epidemic in the US. 

General Comments/Issues:

Comment: If there are ethical or legal restrictions on sharing a de-identified data set, please explain them in detail (e.g., data contain potentially sensitive information, data are owned by a third-party organization, etc.) and who has imposed them (e.g., an ethics committee). Please also provide contact information for a data access committee, ethics committee, or other institutional body to which data requests may be sent… If there are no restrictions, please upload the minimal anonymized data set necessary to replicate your study findings as either Supporting Information files 

Response: I checked with the state funding organization who checked with their legal department. They provided me with approval to upload the de-identified data set. There are no ethical or legal constraints with sharing the data and so I will upload the data set (in Excel format) as a supporting information file. 

Comment: Please include your full ethics statement in the ‘Methods’ section of your manuscript file. In your statement, please include the full name of the IRB or ethics committee who approved or waived your study, as well as whether or not you obtained informed written or verbal consent. If consent was waived for your study, please include this information in your statement as well.

Response: This study received a determination of exemption from human subjects research since the investigators had no interaction with participants for the purpose of collecting data and only secondary analysis of fully de-identified data collected by program staff was done. We already indicated in the full name of the IRB in the methods section of the original manuscript. We have added that written and verbal consent were waived given the exempt status of the study. 

Reviewer #1: 

Comment: Page 18, first sentence "From a practical standpoint, the results indicated that first responders are likely to have greater knowledge of the risks for an opioid-related overdose, signs of an overdose, and what to do to reverse an overdose prior to training compared with first responders" I do apologize but it is unclear for me, first responders compared with first responders???

Response: Thank you for catching this typo. We have amended the sentence to be: “From a practical standpoint, the results indicated that first responders are likely to have greater knowledge of the risks for an opioid-related overdose, signs of an overdose, and what to do to reverse an overdose prior to training compared with bystanders.”

Reviewer #2:

Comment: 1) In table 1, the visualization of the original data must be available to see all the people's choices. The original questionnaire must be included.

Response: On page 8 in the original manuscript, we explain that there were 24 items administered with 2 items removed because nearly 100% of respondents answered them correctly (causing convergence problems since there is almost no variation in the responses). Table 1 showed the 22 items that remained and which were subject to the invariance analyses. We have put back in the 2 items removed for the analyses to reflect the original questionnaire construction. We have identified the removed items and added a sentence to the table note to clarify they were not included in the invariance analysis. But they are now shown in the Table 1 as requested.

Comment: 2) The statistics analysis methods must be more detailed, as much as, the comparison criteria among groups.

Response: Given the level of detail of the ESEM and CFA methods used, we believe this comment pertains to the very brief section that provides sample descriptive statistics in the Study setting and sample section. We agree it would be best if we had more and more detailed sample statistics so that we could, for example, compare the two groups on demographics and other background characteristics. However, as we indicate in the manuscript, the fully anonymous nature of the data collected at the study sites means the data provided did not include demographic or any other information at the individual level. We only know if a training group consisted of first responders or bystanders. The study sites did provide aggregate data on their trainings separately from the OOKS pre-test data. We provided all the aggregate information from the sites, but because of this aggregation, we could not conduct more detailed descriptive statistical comparisons. We also could not subdivide and compare the demographics by bystander subgroup as this information was also not available – a concern we address further below.

Comment: 3) The innovative aspect of the article is not clear. Also, it is not clear the hypothesis of the authors.

Response: We believe the manuscript adds to the literature in two important ways but perhaps we were not as clear about these as we could have been. We have added the following text (on page 5 of our revised manuscript and at the end of the introduction section): 

To the best of our knowledge, there has not been a measurement invariance analysis of the OOKS to test whether it measures background knowledge for these two groups for whom naloxone education and training are commonly conducted. To address this gap, we conducted a multigroup analysis of OOKS item equivalence by assessing levels of measurement invariance (i.e., configural, metric, and scalar) for first responders and bystanders. 

We also believe that the application of ESEM, a relatively new and not widely used technique for analyzing the factor structure of measure items is a further innovative contribution. Immediately after the added text, the original manuscript continues: 

We also wanted to re-examine the OOKS item factor structure using a relatively new analytic tool, exploratory structural equation modeling (ESEM), which combines elements of exploratory factor analysis and confirmatory factor analysis. We describe the potential advantages of ESEM in the analyses section below.

We did not have a specific hypothesis but instead were guided by a research question: Does the OOKS perform equally well assessing knowledge of signs of an overdose and naloxone administration by first responders and bystanders? To clarify this point, we added this text immediately following the sentences supporting the study’s innovativeness:

If the OOKS demonstrates measurement invariance then one version would be suitable for use with both groups whereas lack of measurement invariance suggests either different versions of the OOKS containing different item subsets or different scoring thresholds for bystanders and first responders would be more appropriate.

We hope these additions provides adequate support for the study’s innovativeness and unique contribution made as well as the intent of the study to assess the OOKS measurement invariance and the implications of having or not being invariant. 

Comment: 4) On page 18 the authors state: "Despite these results, and for several reasons.... What are the several reasons mentioned?

Response: The several reasons are enumerated after that statement. Below is the paragraph in which this statement occurs, provided for context. We added the clause, as enumerated below, to clarify that the reasons for selecting the 3-factor model follow. We also added a reference to support the assertion that a factor was identified only when 3 or more items has loadings greater than .50 to further support the model selected.

…Despite these results, and for several reasons enumerated below, we selected between the 3- (RMSEA=.044 [95% CI=.041 - .048]; CFI=.941; TLI=.919; SRMR=.067) and 4-factor models (RMSEA=.044 [95% CI=.034 - .041]; CFI=.962; TLI=.941; SRMR=.057), ultimately selecting the 3-factor model for further analysis. 

As noted by Finch (2020) in his simulation study, when the fit statistics are inconsistent with the underlying factor structure of the simulated models, the statistics tend to favor over-factored (i.e., too many factors) results. Parsimony was also an important consideration. All else being equal, simpler models are better given relatively similar fit statistics. One of the factors in the 4-factor model had only two items with loadings greater than .50, with the 5 and 6-factor models having a similar factor structure whereby only several items had loadings greater than .50. Current recommendations are that a factor is identified when there are 3 or more items with sufficient loadings.(38) Finally, in subsequent preliminary analyses, the 4-factor model failed to converge at the metric invariance step suggesting it could be overly complex. Given that both the 3- and 4-factor model fit statistics indicated both fit the data well and the item to factor structure appeared to be more robust in the 3-factor model, which had more than 2 items with loadings > .50 on every factor, and the pattern of factor loadings for the items made sense substantively, we selected the 3-factor model for measurement invariance testing. 

Comment: 5) There is a lack of literature discussion. The authors do not compare original data with general knowledge. The factors included and the importance of each must be contextualized.

Response: Thank you for pointing out this oversight. We did not discuss the factor structure we obtained and how it did or did not comport with the original instrument’s structure as intended or of the structure found in another study to develop a much shorter version of the OOKS that is called the BOOK. We have added this additional language in the discussion to better put our factor structure findings into context:

…The original OOKS was organized into four sections based on substantive considerations that were not derived through statistical analyses.(18) The four sections include: (1) questions on knowledge of risks for an overdose; (2) signs of an overdose; (3) actions to take in response to an overdose; and (4) the correct use of naloxone. Our 3-factor structure more closely approximates the BOOK factor structure whereby, essentially, actions to take in response to an opioid-related overdose and the correct use of naloxone form a single rather than two separate factors. We believe the similarity of the factor analytic results across our study and the BOOK development study as well as the general consistency of both factor-analytic studies with the substantively driven design of the original OOKS suggests the 3-factor structure we identified is robust and well captures the knowledge areas that should be covered during OEND trainings. 

Comment: 6) The results are not overall representative to make a general statement of it.

Response: We believe we were careful to point out that our sample was not generalizable if that is what is meant by overall representative. For example, on page 19 we state the following: 

Generalizability of study findings remains a potential issue. Although we had a large and diverse sample, participants were drawn from two counties in a single midwestern state in the United States and might not be representative of OEND training participants in other parts of the country or in other countries or with other backgrounds.

We do stand by the finding of measurement invariance for bystanders and first responders meaning the selected items worked equally well as a test of knowledge for both groups. And we tried not to make claims that this finding would hold universally for different kinds of OEND training participants despite having a large and (moderately) diverse sample collected from two sites and across many training sessions. We believe we do present the findings as being qualified with respect to generalizability and state we do not have complete confidence in the generalizability of the findings:

From a practical standpoint, the results indicated that first responders are likely to have greater knowledge of the risks for an opioid-related overdose, signs of an overdose, and what to do to reverse an overdose prior to training compared with bystanders. That this finding held across different groups of bystanders trained in two different counties, with considerable heterogeneity within each bystander group, gives us some though not complete confidence in the generalizability of the findings.

If there is a specific statement where we have made overly broad claims for the findings, however, we would be happy to review and amend as needed. 

Comment: The bystander group of analysis should be divided into drug users, family members, and professionals. The way the authors represent does not point to the right source of misinformation. It is not clear where/who the questionnaire requires modification.

Response: We are not certain as to where we indicate the questionnaire requires modification. Our analyses indicated that the form has full measurement invariance for bystanders (as a group) and first responders. This means as a pre-test for our sample, the selected OOKS items identified gaps in knowledge as well as strengths in knowledge about equally well for both groups and that mean comparisons of knowledge gain post-training should be valid for either group. We did identify that, collectively, bystanders were less knowledgeable than first responders and indicated the specific items where knowledge was most lacking. For example, on page 20 of the original manuscript, we state the following in the conclusions:

Although not a focus of the study, we also found that bystanders tend to have larger knowledge gaps in key areas related to recognizing and reversing an opioid overdose, particularly with respect to details on naloxone use and duration of effect. These areas could be more emphasized in future trainings to address these knowledge gaps. 

We understand regarding there being no further analysis of subgroups to identify the source of misinformation if what is meant by misinformation is that the lower background knowledge of the bystanders. We do indicate in the limitations that we were unable to disentangle the results for these bystander subsamples but have added these statements in the limitations section (page 19 of the track-changed manuscript) to provide what we hope is further clarity as to why further analysis was not possible and the implications of that: 

Given the data were collected to fully preserve participant anonymity, we could not subdivide this sample to compare these different subpopulations for OEND knowledge and knowledge gaps. Consequently, we were unable to determine if gaps in knowledge among bystanders were differentially attributable to drug users, family members (of drug users), or professionals working in treatment settings Future research should examine whether this shortened version of the OOKs works equally well for these potentially different subgroups of bystanders.

Thank you again for providing the thoughtful reviews and for the opportunity to revise and resubmit. We think the manuscript has been strengthened because of the revisions made in response to the reviews. We hope these revisions adequately address all the reviewers’ concerns and/or that our responses explain why some revisions are not feasible. But we are prepared to make further changes if requested. We look forward to hearing a decision on the revised manuscript.

Sincerely, 

James A. Swartz, Ph.D.

Professor

University of Illinois Chicago

Jane Addams College of Social Work

---

## [Editor Report · Decision Letter 1]

27 Sep 2022

A measurement invariance analysis of selected opioid overdose knowledge scale (OOKS) items among bystanders and first responders

PONE-D-22-18050R1

Dear Dr. Swartz,

We’re pleased to inform you that your manuscript has been judged scientifically suitable for publication and will be formally accepted for publication once it meets all outstanding technical requirements.

Kind regards,

Michelle Melgarejo da Rosa

Academic Editor

PLOS ONE
---

## [Editor Report · Acceptance letter]

5 Oct 2022

PONE-D-22-18050R1 

A measurement invariance analysis of selected Opioid Overdose Knowledge Scale (OOKS) items among bystanders and first responders 

Dear Dr. Swartz:

I'm pleased to inform you that your manuscript has been deemed suitable for publication in PLOS ONE. Congratulations! Your manuscript is now with our production department. 

Kind regards, 

on behalf of

Dr. Michelle Melgarejo da Rosa 

Academic Editor

PLOS ONE